# NILMPEds: A Performance Evaluation Dataset for Event Detection Algorithms in Non-Intrusive Load Monitoring

**Lucas Pereira** [1,2] 

1   ITI, LARSyS, 9020-105 Funchal, Portugal; lucas.pereira@tecnico.ulisboa.pt
2   Ténico Lisboa, Universidade de Lisboa, 1049-001 Lisbon, Portugal

**Abstract:** Datasets are important for researchers to build models and test how these perform, as well as to reproduce research experiments from others. This data paper presents the NILM Performance Evaluation dataset (NILMPEds), which is aimed primarily at research reproducibility in the field of Non-intrusive load monitoring. This initial release of NILMPEds is dedicated to event detection algorithms and is comprised of ground-truth data for four test datasets, the specification of 47,950 event detection models, the power events returned by each model in the four test datasets, and the performance of each individual model according to 31 performance metrics.

**Keywords:** dataset; performance evaluation; performance metrics; event detection; non-intrusive load monitoring; disaggregation; NILM; smart grid

---

## 1. Summary

Public datasets are crucial elements in data science research as these not only allow researchers to perform systematic evaluations and benchmarks of their algorithms but also enable other researchers to replicate and reproduce existing research [1].

Non-Intrusive Load Monitoring (NILM or load disaggregation) is the process of estimating the energy consumption of individual appliances from electric power measurements taken at a limited number of locations in the electrical distribution of a building [2]. A typical NILM dataset is a collection of electrical energy measurements, taken from the mains (i.e., aggregate consumption) and from the individual loads (i.e., ground-truth data, which are obtained either by measuring each load at the plug-level or measuring the circuit to which the load is connected [3].

As presented in a recent review [3], there are over 20 public datasets for NILM research. According to the same review, these datasets can be categorized according to their suitability to be used to evaluate event-based and event-less approaches [4]. Event-based strategies seek to disaggregate the total consumption employing detecting and labeling appliance transition (referred to as power events) in the aggregated signal. As such, datasets for event-based NILM must also include labels for the power events from the appliances of interest. On the other hand, event-less approaches attempt to match each sample of the aggregated power to the consumption of one specific device, or a combination of different devices. Therefore, datasets for event-less approaches do not require any labeled transitions.

Existing NILM datasets are also categorized according to the data reporting rates [5]: macroscopic datasets with data reporting rates around 1 Hz, and microscopic datasets with rates of several kHz.

The Building-Level fUlly-labeled dataset for Electricity Disaggregation (BLUED) [6] and the Energy Monitoring through Building Electricity Disaggregation (EMBED) [7] are examples of microscopic datasets for event-based NILM. The Almanac of Minutely Power dataset (AMPds) [8], REFIT (Personalised Retrofit Decision Support Tools For UK Homes Using Smart Home Technology) Project Electrical Load Measurements Dataset [9], and the Rainforest Automation Energy Dataset for Smart Grid Meter Data Analysis (RAE) [10] are examples of macroscopic datasets for event-less approaches. Finally, UK-DALE [11] is an example of an event-less dataset that fits both microscopic and macroscopic data rates, as it contains both high- and low-frequency data for three out of five houses [3].

This data descriptor presents NILMPEds (NILM Performance Evaluation dataset). NILMPEds is a different type of NILM dataset, in a sense that it is aimed primarily at research reproducibility concerning the development and performance evaluation of event detection algorithms. Event detection is the process of identifying the relevant changes (i.e., appliance) in the aggregate consumption data [2]. NILMPEds contains the results of 47,950 event detection models when applied to four public event detection datasets. The different parameter configuration of each model and the ground-truth data are also available. Finally, this dataset also contains the performance evaluation of each model according to 31 performance metrics.

The data in NILMPEds were initially collected to serve as baseline data to study the behavior of performance metrics for event detection algorithms [12]. This is a topic of particular interest for the NILM research community, given that there is no consensus concerning which performance metrics to employ in NILM performance evaluation [13,14].

Nevertheless, NILMPEds is also suitable to benchmark high- and low-frequency event detection algorithms (e.g., [15]). Furthermore, the different detection models can be used to test data labeling platforms [16–18], as well as to study the potential impacts of event detection algorithms in smart-grid applications that require this information (e.g., appliance activation forecast for smart-charging of electric vehicles and battery energy storage systems).

Despite the obvious differences with the other NILM datasets, since NILMPEds derives from two microscopic datasets and five event detection algorithms (details in Section 2.2), it should be considered under the category of microscopic event-based datasets.

## 2. Methods

### 2.1. Event Detection Dataset

The event detection results presented in this paper were obtained from the BLUED and the UK-DALE datasets, more concretely Phases A and B of BLUED, and one week of data from UK-DALE's Houses 1 and 2.

The BLUED dataset consists of load demand data from one house in the USA with a two-phase electric installation. The voltage and current sampled at 12 kHz is available, as well as a list of all the power events with an absolute power change of at least 30 Watts. The UK-DALE dataset consists of power demand data from five houses in the UK. For three of the five houses (Houses 1, 2 and 5), the dataset also contains the whole-house voltage and current sampled at 16 kHz.

Before using the datasets, it was necessary to compute the active and reactive power from the high-frequency voltage and current measurements. These values were computed at line frequency, i.e., 60 Hz for BLUED and 50 Hz for UK-DALE. Furthermore, since no individual labels were available for UK-DALE, this had to be done manually. The data labeling was done three steps: (1) an expert heuristic detector was applied to the active power at 50 Hz; (2) the obtained false positives were removed manually; and (3) the missing labels (i.e., false negatives) were added manually. It is important to remark that, for the sake of consistency with BLUED, only power events with a minimum absolute power change of 30 Watts were labeled. For additional details about this process, please refer to [17].

Table 1 summarizes these datasets. **Rate** is the sampling rate at which active and reactive power was calculated; **P.E.** is the number of power events in the dataset; **Power Change (W)** is a summary

of distribution of the power events in terms of mean, and the 25%, 50%, and 75% percentiles; and **Elapsed Time (s)** is a summary of the difference in seconds between the power events in the same terms as the Power Change column.

Please note that only the ground-truth data for these datasets are made available in NILMPEds (see Section 3.1). However, the interested readers should refer to [19] for additional details and download information.

**Table 1.** Summary of the active power change and elapsed time between power events in the event detection datasets.

| Dataset Name | Dataset ID | Rate | P.E. | Power Change (W) | | | | Elapsed Time (s) | | | |
|---|---|---|---|---|---|---|---|---|---|---|---|
| | | | | Mean | 25% | 50% | 75% | Mean | 25% | 50% | 75% |
| UK-DALE H1 | 1 | 50 Hz | 5440 | 268 | 48 | 100 | 273 | 111 | 4 | 7 | 28 |
| UK-DALE H2 | 2 | 50 Hz | 2842 | 365 | 45 | 74 | 137 | 212 | 6 | 15 | 172 |
| BLUED PA | 3 | 60 Hz | 887 | 274 | 84 | 116 | 582 | 690 | 18 | 294 | 892 |
| BLUED PB | 4 | 60 Hz | 1562 | 351 | 40 | 170 | 428 | 383 | 7 | 35 | 83 |

### 2.2. Event Detection Algorithms and Models

The event detection models are based on five different event detection algorithms, one expert heuristic, and four probabilistic detectors. For each algorithm, the event detection models were obtained by means of a parameter sweep, i.e., a controlled variation of a number of parameters in each algorithm. Ultimately, the parameter sweep returned 47,950 distinct event detection models across the five algorithms. The five algorithms and the respective number of models are presented in Table 2.

**Table 2.** Number of different models evaluated across the four datasets.

| Event Detection Algorithm | | Models | Model-Dataset Pairs |
|---|---|---|---|
| Name | ID | | |
| Simplified LLR Detector with Maxima [15] | 1 | 1 k | 4 k |
| LLR Detector with Maxima [15] | 2 | 1 k | 4 k |
| Simplified LLR Detector with Voting [15] | 3 | 1.1 k (50 Hz); 9.5 k (60 Hz) | 22 k; 19 k |
| LLR Detector with Voting [20] | 4 | 1.1 k (50 Hz); 9.5 k (60 Hz) | 22 k; 19 k |
| Expert Heuristic Detector [21] | 5 | 4.95 k | 19.8 k |
| | | **47.95 k** | **109.8 k** |

Extensive details about the event detection algorithms and the parameter sweep are out of the scope of this paper, but the interested readers can refer to [15], and the respective algorithm publications for more information. NILMPEds, instead, makes available the values of the different parameters for each detection model (see Section 3.2).

### 2.3. Power Events and Performance Metrics

Each detection model was trained and tested against the four datasets for a total of 109,800 detection model/dataset pairs. From these, 31 performance metrics were calculated. For more details about the performance metrics, and their calculation, the interested readers should refer to [22].

The performance of each model was calculated taking into consideration a tolerance interval, $\Omega$, in which the detected power events should fall to be considered correct detections. The detection interval is defined by Equation (1). It is based on the ground-truth position (GT), and a tolerance (Tol) value (in samples) that was set to account for eventual ambiguity when defining exactly where an event occurs during the ground-truth labeling process.

$$\Omega = [GT - Tol, GT + Tol] \tag{1}$$

In previous work on this topic [23], the authors varied this parameter from 1 to 6 seconds (in 1-second steps) and found that there were no improvements with more than 3 seconds. Consequently, it was decided to set this parameter to range between 0 and 3 seconds with variable steps, as defined by the set $\tau$ in Equation (2), where $F_s$ is the sampling rate of the dataset. Ultimately, this results in 10 values per metric for each detection model/dataset pair.

$$\tau = \{0, 1, 5, 15, F_s, 1.5 \times F_s, 2 \times F_s, 2.5 \times F_s, 3 \times F_s\} \tag{2}$$

## 3. Data Description

NILMPEds is made available in four different folders, and all data files are in *Comma Separated Values* format (CSV). Figure 1 provides an overview of the underlying organization of NILMPEds. The following subsections thoroughly describe the content of the different folders.

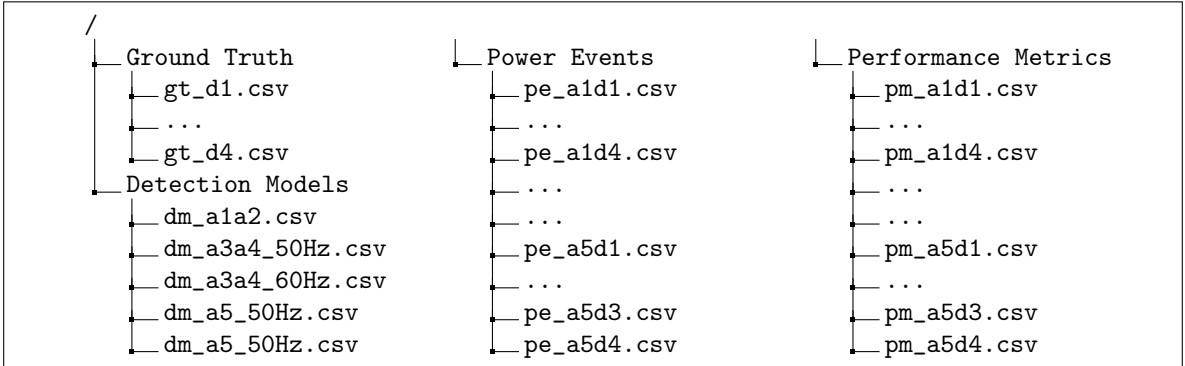

**Figure 1.** Underlying folder and file organization of NILMPEds.

### 3.1. Ground Truth

This folder contains the ground truth data that were used to evaluate the event detection models, one CSV file per dataset. The underlying fields are described in Table 3.

**Table 3.** Column descriptions for the ground truth files (gt_d<?>.csv).

| Column | Description |
| --- | --- |
| *Position* | Position of the power event in number of samples from the beginning of the dataset. |
| *Delta_P* | Difference in power before and after the power event, considering the average of one second of samples before and after the event position. |
| *Distance* | Distance in samples to the previous power event. |
| *Day* | The corresponding day in the dataset. The days are numbered from 1 to the total number of days in the dataset. |

### 3.2. Event Detection Models

This folder contains the values for the parameters for each of the 47,950 event detection models. Table 4 describes the underlying fields.

**Table 4.** Column descriptions for the event detection model files (dm_*.csv).

| Column | Description |
|--------|-------------|
| **All Models (dm_*.csv)** | |
| *Model_ID* | Unique model identifier for each event detection algorithm. |
| *w0* | Length of the pre-event window in seconds. |
| *w1* | Length of the post-event window in seconds. |
| **Models for Algorithms 1 and 2 (dm_a1a2.csv)** | |
| *Mpre* | Maxima precision in seconds. |
| **Models for Algorithms 3 and 4 (dm_a3a4_<?>Hz.csv)** | |
| *wV* | Length of the voting window in seconds. |
| *Vthr* | Minimum number of votes necessary to trigger a power event. |
| **Models for Algorithm 5 (dm_a5_<?>Hz.csv)** | |
| *Gpre* | Number of seconds before the second under evaluation |
| *tElap* | Minimum elapsed time between events |
| *Eedge* | Sample index inside the second where the event occurred |

## 3.3. Power Events

This folder contains data for the power events that were detected by each of the 109,800 detection model/dataset pairs. The power events are grouped by algorithm and dataset, in a total of 20 CSV files (5 algorithms × 4 datasets). The underlying fields in each file are described in Table 5.

**Table 5.** Column descriptions for the power event files (pe_a<?>d<?>.csv).

| Column | Description |
|--------|-------------|
| *ID* | Power event unique identifier. |
| *Model_ID* | Model identifier. This corresponds to the Model_ID in the *detection models* data. |
| *Position* | Position of the power event in number of samples from the beginning of the dataset. |
| *Delta_P* | Difference in power before and after the power event, considering the average of one second of samples before and after the event position. |
| *ds* | Value of the detection statistics. This field is not available for the models of Algorithm 5 since these are not probabilistic. |

## 3.4. Performance Metrics

This folder contains the performance data for each of the 109,800 detection model/dataset pairs. The performance metrics are also grouped by algorithm and dataset, in a total of 20 CSV files. The underlying fields in each file are described in Table 6, where **Best** and **Worst** refer to the theoretical best and worst values of each metric, respectively.

**Table 6.** Column descriptions for the performance metric files (pm_a<?>d<?>.csv).

| Column | Description | Values Range Best | Values Range Worst |
|--------|-------------|------|-------|
| *Model_ID* | Model identifier. This corresponds to the Model_ID in the *power events* data. | - | - |
| *Tolerance* | A tolerance value (in samples) that was set to account for eventual ambiguity when labeling the event detection datasets. | - | - |
| *Events* | The number of real power events in the dataset. | - | - |
| *TP* | True Positives | Events | 0 |
| *FP* | False Positives | 0 | S-Events [a] |
| *TN* | True Negatives | S-Events [a] | 0 |
| *FN* | False Negatives | 0 | Events |
| *A* | Accuracy | 1 | 0 |
| *E* | Error-rate | 0 | 1 |
| *P* | Precision | 1 | 0 |

**Table 6.** *Cont.*

| Column | Description | Values Range Best | Worst |
|--------|-------------|------|-------|
| *R* | Recall | 1 | 0 |
| *FPR* | False Positive Rate | 0 | 1 |
| *TPP* | True Positive Percentage | 1 | 0 |
| *FPP* | False Positive Percentage | 0 | [a] |
| *TNR* | True Negative Rate | 1 | 0 |
| *FDR* | False Discovery Rate | 0 | 1 |
| *F05* | $F_{0.5}$-Score | 1 | 0 |
| *F1* | $F_1$-Score | 1 | 0 |
| *F2* | $F_2$-Score | 1 | 0 |
| *MCC* | Matthews Correlation Coefficient | 1 | −1 |
| *SMCC* | Standardized MCC | 1 | 0 |
| *DPS_PR* | Distance to Perfect Score between Precision and Recall | 0 | 2 |
| *DPS_Rate* | Distance to Perfect Score between TPR (i.e., Recall) and FPR | 0 | [b] |
| *DPS_Perc* | Distance to Perfect Score between TPP and FPP. | 0 | [b] |
| *WAUC* | Wilcoxon statistics based Area Under Curve | 1 | 0 |
| *WAUCB* | Wilcoxon statistics based AUC Balanced | 1 | 0 |
| *GAUC* | Geometric Mean AUC | 1 | 0 |
| *BAUC* | Biased AUC | 1 | 0 |
| *TPC_FP* | Total Power Change - False Positives | 0 | [c] |
| *TPC_FN* | Total Power Change - False Negatives | 0 | [c] |
| *APC_FP* | Average Power Change - False Positives | 0 | [c] |
| *APC_FN* | Average Power Change - False Negatives | 0 | [c] |
| *DPS_TPC* | Distance to Perfect Score TPC | 0 | [d] |
| *DPS_APC* | Distance to Perfect Score APC | 0 | [d] |

[a] *S* is the number of samples in the event detection dataset. [b] Since the FPP metric can return a value greater than 1, it is not possible to define a fixed lower bound. [c] The worst result is proportional to the number of events and size of the erroneous events; thus, it is not possible to define a fixed lower bound. [d] Since we cannot define a fixed lower bound to the individual metrics, it is also not possible to set a lower bound to the DTP metric.

## 4. Data Exploration

As an application example, Figure 2 shows the distribution of event detections across the five algorithms and four datasets. The horizontal dotted line in each graph represents the number of actual power events in the dataset. As can be observed, algorithms one and two are more conservative when compared to the other alternatives. It is also evident that Algorithms 3 and 4 detect much more events than the other three options. The voting threshold parameter in part explains this effect, since setting it to a low value (e.g., 5) will result in very liberal event detection models.

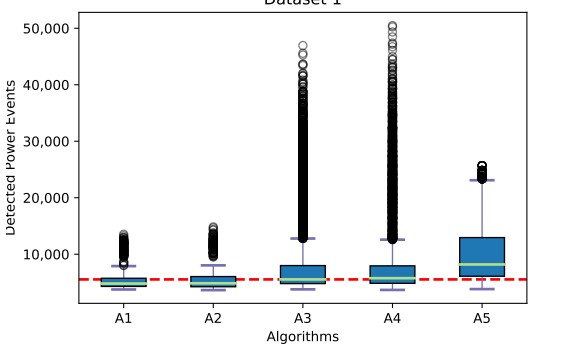 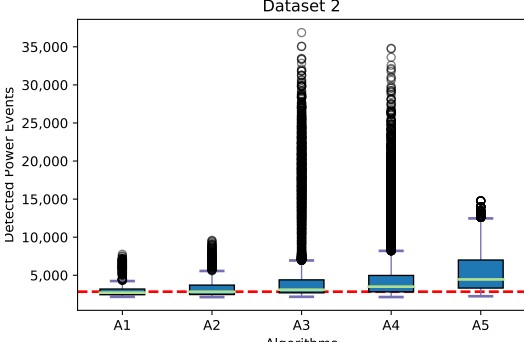

**Figure 2.** *Cont.*

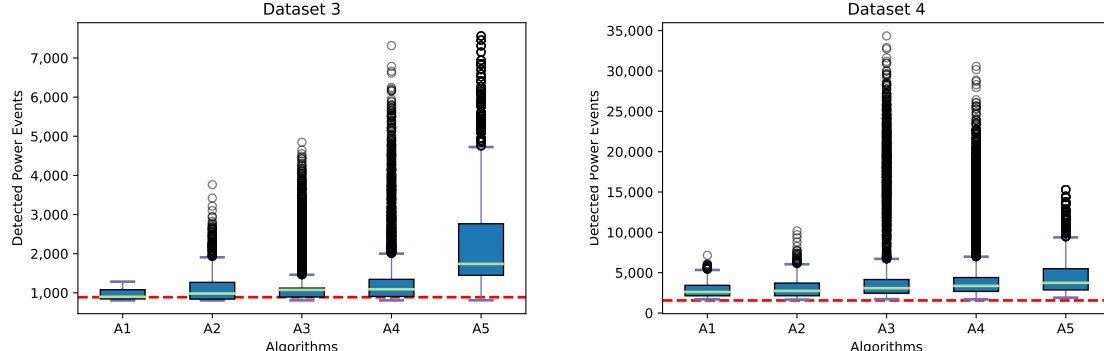

**Figure 2.** Distribution of detected power events across the five algorithms and four datasets.

*Performance Metrics*

Another application example is provided in Figure 3 that shows the median $F_1$-score for each algorithm across the four datasets. As it can be observed, after a tolerance value of *Fs* (i.e., $\pm 50$ or 60 samples), the detection results are not significantly different. In contrast, tolerance values below $\pm 5$ samples lead to particularly poor outcomes. Finally, it is also evident from the results that the heuristic detector (Algorithm 5) tends to be outperformed by its probabilistic counterparts.

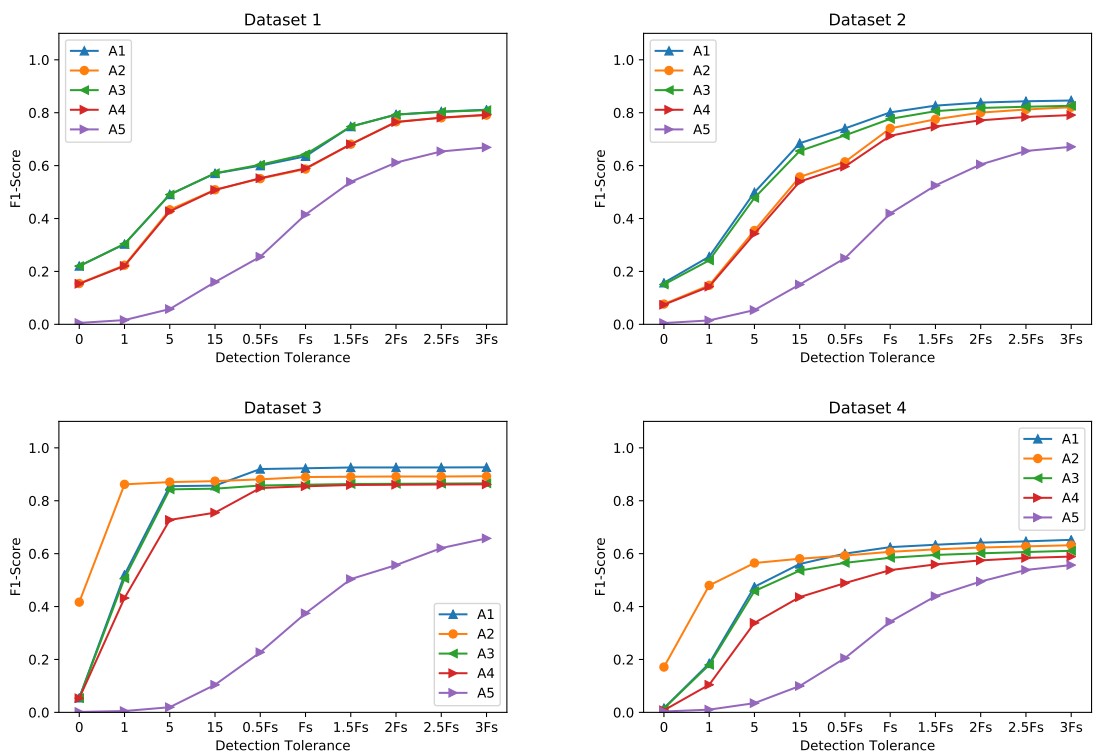

**Figure 3.** Median $F_1$-Score, for each dataset, under the different detection tolerance values.

## 5. Future Work and Source Code Release

This data descriptor presents NILMPEds, a performance evaluation dataset for event detection algorithms in NILM.

One limitation of the current version of NILMPEds is the low number of detection algorithms and public datasets used in the evaluation process. Consequently, to increase its value, a future version will also include results from the EMBED dataset. EMBED is one of the few NILM datasets that contain labeled power events by default. Furthermore, to promote the addition of new event detection algorithms,

the source code used to generate NILMPEds will be released along with the dataset, in the Open Science Framework online platform[1].

**Funding:** This research was funded by the Portuguese Foundation for Science and Technology (FCT) under projects UID/EEA/50009/2013, UID/EEA/50009/2019, and CEECIND/01179/2017.

**Acknowledgments:** The author thanks the anonymous reviewers for their constructive comments.

**Conflicts of Interest:** The author declares no conflict of interest.

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
