# Peer review of "NILMPEds: A Performance Evaluation Dataset for Event Detection Algorithms in Non-Intrusive Load Monitoring"

_data, 2019_

Round 1

Reviewer 1 Report

The authors are suggested to improve the paper in the following ways:

- does this dataset target high frequency NILM or low frequency NILM?

- the 5 event detection algorithms are far from the state-of-the-art in this field, should include some latest development of NILM

- should show some sample of how you rearrange the public datasets of UKDALE and BLUED. as far as I know, UKDALE only contains low-frequency data.

- from the current version of the manuscript, I suspect that this work can promote the benchmarking across different NILM solutions.

Author Response

Thank you very much for your comments.

Please find our responses in the attached file.

Kind regards,

Reviewer 2 Report

I have 2 main issues with this manuscript that requires major revisions:

(1) the author cites himself, too much. Why not cite other seminal papers, such as: https://link.springer.com/article/10.1007/s12053-014-9306-2

(2) author uses only 2 datasets. There are other more prominent one such as REFIT and AMPds.

Author Response

(The authors gave the same response as above.)

Reviewer 3 Report

The paper deals with a dataset for the detection of power consumption events. The Authors are presenting in the paper how the data in the data set has been organized, the parameters of event detection models and the considered performance metrics.

The Authors should provide in the final version of the paper some clarification. Does the “Freq.” field of the table 1, refer to the frequency of the line or to the data sample?

In addition, the Authors should describe some details about the two public NILM datasets, to ensure the self-consistency of the paper. A short description of the public data sets is more than enough.

Author Response

(The authors gave the same response as above.)

Round 2

Reviewer 2 Report

The authors have address all of my concerns in their edits, and have given appropriate explanations in their reply letter. This manuscript is now stronger and is, in my opinion, publishable.